# Dipolar Order Mediated $^1$H$\rightarrow$$^{13}$C Cross-Polarization for Dissolution-Dynamic Nuclear Polarization

Stuart J. Elliott[1], Samuel F. Cousin[1], Quentin Chappuis[1], Olivier Cala[1], Morgan Ceillier[1], Aurélien Bornet[2], Sami Jannin[1]

[1]Centre de Résonance Magnétique Nucléaire à Très Hauts Champs - FRE 2034 Université de Lyon / CNRS / Université Claude Bernard Lyon 1 / ENS de Lyon, 5 Rue de la Doua, 69100 Villeurbanne, France
[2]Institut des Sciences et Ingénierie Chimiques, Ecole Polytechnique Fédérale de Lausanne (EPFL), Batochime, CH-1015 Lausanne, Switzerland

*Correspondence to*: Stuart J. Elliott (stuart-james.elliott@univ-lyon1.fr)

**Abstract.** Magnetic resonance imaging and spectroscopy often suffer from a low intrinsic sensitivity, which can in some cases be circumvented by the use of hyperpolarization techniques. Dissolution-dynamic nuclear polarization offers a way of hyperpolarizing $^{13}$C spins in small molecules, enhancing their sensitivity by up to four orders of magnitude. This is usually performed by direct $^{13}$C polarization, which is straightforward but often takes more than an hour. Alternatively, indirect $^1$H polarization followed by $^1$H$\rightarrow$$^{13}$C polarization transfer can be implemented, which is more efficient and faster but is technically very challenging and hardly implemented in practice. Here we propose to remove the main roadblocks of the $^1$H$\rightarrow$$^{13}$C polarization transfer process by using alternative schemes with: (*i*) less *rf*-power; (*ii*) less overall *rf*-energy; (*iii*) simple *rf*-pulse shapes; and (*iv*) no synchronized $^1$H and $^{13}$C *rf*-irradiation. An experimental demonstration of such a simple $^1$H$\rightarrow$$^{13}$C polarization transfer technique is presented for the case of [1-$^{13}$C]sodium acetate, and is compared with the most sophisticated cross-polarization schemes. A polarization transfer efficiency of ~0.43 with respect to cross-polarization was realized, which resulted in a $^{13}$C polarization of ~8.7% after ~10 minutes of microwave irradiation and a single polarization transfer step.

## 1 Introduction

Traditional magnetic resonance imaging (MRI) and spectroscopy (MRS) experiments usually suffer from low sensitivity. Hyperpolarization techniques including dissolution-dynamic nuclear polarization (*d*DNP) can be used to highly polarize a large variety of chemical systems and therefore enhance nuclear magnetic resonance (NMR) signals by several orders of magnitude (Ardenkjær-Larsen et al., 2003). Various applications of *d*DNP have been demonstrated including the study of enzyme kinetics, cell extracts and heteronuclear metabolomics (Bornet et al., 2014; Dumez et al., 2015; Bornet et al., 2016). Most *d*DNP applications involve the use of weakly magnetic isotopes such as $^{13}$C, but excessively long DNP timescales $\tau_{DNP}(^{13}C)$ hinder efficient polarization build-up and lead to extended experimental times. Intrinsically sensitive proton nuclear spins do not suffer from such issues and can be polarized quickly and to a greater extent at low temperatures (Hartmann et al., 1973).

The use and optimization of cross-polarization (CP) under *d*DNP conditions (typically at temperatures of about 1.2-1.6 K in superfluid helium) provides a way to substantially boost $^{13}$C polarizations and enhance build-up rates $1/\tau_{DNP}(^{13}C)$ (by a factor of up to 40) (Hartmann and Hahn, 1962; Pines et al., 1972; Perez Linde, 2009; Jannin et al., 2011; Bornet et al., 2012; Batel et al., 2012; Bornet et al., 2013; Vuichoud et al., 2016; Cavaillès et al., 2018). The technique requires intense $B_1$-matching (typically > 15 kHz) of simultaneous $^1$H and $^{13}$C spin-locking radiofrequency (*rf*) fields throughout an optimized contact period (typically > 1 ms). This CP-DNP approach recently turned out to be key for the preparation of transportable hyperpolarization (Ji et al., 2017) where samples are polarized in a CP equipped polarizer and then transported over extended periods (typically hours or days) to the point of use.

This CP approach has been demonstrated on typical *d*DNP samples back in 2012 (Bornet et al., 2012), however, the technique remains challenging today because of its methodological and technical complexity. Indeed, CP under *d*DNP conditions employs

sophisticated pulse sequences, and involves high power and energy *rf*-pulses. Another drawback of CP-DNP is that it can hardly
be scaled-up to volumes larger than 500 $\mu$L, otherwise engendering detrimental arcing in the superfluid helium bath (Vinther et al.,
2019). Such scaling-up would be required for enabling parallel hyperpolarization of multiple transportable samples (Lipsø et al.,
2017), and for volumes >1 mL currently used for hyperpolarized human imaging (Nelson et al., 2013).
For hyperpolarizing larger sample volumes, alternative *rf*-sequences with reduced power requirements are desired. Lower
power alternatives to CP have previously been described in the literature (Jeener et al., 1965; Jeener and Broekaert, 1967; Redfield,
1969; Kunitomo et al., 1974; Demco et al., 1975; Emid et al., 1980; Vieth and Yannoni, 1993; Zhang et al., 1993; Kurur and
Bodenhausen, 1995; Lee and Khitrin, 2008; Khitrin et al., 2011; Vinther et al., 2019), which rely on indirect polarization transfer
via proton dipolar order rather than through a direct $^1$H-$^{13}$C Hartman-Hahn matching condition (Hartmann and Hahn, 1962).
The population of a Zeeman eigenstate for a spin-1/2 nucleus at thermal equilibrium $\rho_{eq}^i$ is given as follows:

$$\rho_{eq}^i = \frac{\exp\left\{-\frac{\hbar\omega_i}{\kappa_B T}\right\}}{Z}, \tag{1}$$


where $\omega_i$ is the energy of the state for the spin of interest, $T$ is the temperature and $Z$ is a canonical partition function. In the high-
temperature limit, the spin density operator $\hat{\rho}_{eq}$ (which describes the state of an entire ensemble of spin-1/2 nuclei at thermal
equilibrium) is expressed by using a truncated Taylor series:

$$\hat{\rho}_{eq} \simeq \hat{1} + \mathbb{B}\sum_i \hat{I}_{iz}, \tag{2}$$


where $\mathbb{B} = \hbar\omega_0/\kappa_B T$, $\omega_0$ is the nuclear Larmor frequency for the spins of interest and $\hat{I}_{iz}$ is $z$-angular momentum operator for
spin $i$. The second term in Equation 2 corresponds to longitudinal magnetization. However, outside of the high-temperature
approximation higher order terms in the spin density operator expansion cannot be ignored:

$$\hat{\rho}_{eq} \simeq \hat{1} + \mathbb{B}\sum_i \hat{I}_{iz} + \frac{\mathbb{B}^2}{2}\sum_i \sum_j \hat{I}_{iz} \cdot \hat{I}_{jz}. \tag{3}$$


The third term in Equation 3 reveals the presence of nuclear dipolar order (Fukushima and Roeder, 1981) which can in principle
be prepared by generating strongly polarized spin systems, such as those established through conducting *d*DNP experiments
(Sugishita et al., 2019). Such dipolar order can also be efficiently generated by suitable *rf*-pulse sequences, and ultimately used to
transfer polarization (Jeener et al., 1965; Jeener and Broekaert, 1967; Redfield, 1969; Kunitomo et al., 1974; Demco et al., 1975;
Emid et al., 1980; Vieth and Yannoni, 1993; Zhang et al., 1993; Kurur and Bodenhausen, 1995; Lee and Khitrin, 2008; Khitrin et
al., 2011; Vinther et al., 2019). For the sake of simplicity, we will refer here to such polarization transfer schemes as *d*CP for dipolar
order-mediated cross-polarization.
In this Article, we revisit the concept of $^1$H→$^{13}$C *d*CP polarization transfer and assess its efficiency in the context of *d*DNP
experiments at 1.2 K and 7.05 T. We show that for a sample of [1-$^{13}$C]sodium acetate, a $^{13}$C polarization of ~8.7% can be achieved
after ~10 minutes of $^1$H DNP and the use of a sole polarization transfer step. The overall *d*CP transfer efficiency is ~0.43 with
respect to the most sophisticated and efficient high power CP sequences available today. The experimental data presented indicate
that $^1$H Zeeman order ($\hat{I}_z$) is first converted to $^1$H-$^1$H dipolar order ($\hat{I}_{1z} \cdot \hat{I}_{2z}$) and presumably subsequently converted to the desired
$^{13}$C Zeeman order ($\hat{S}_z$). We show how the use of microwave gating (Bornet et al., 2016) is key to *d*CP as it improves the overall
efficiency by a factor more than ~2.3.

**2 Methods**

## 2.1 Sample Preparation and Freezing

A solution of 3 M [1-$^{13}$C]sodium acetate in the glass-forming mixture $H_2O$:$D_2O$:glycerol-$d_8$ (10%:30%:60% v/v/w) was doped with 50 mM TEMPOL radical (all compounds purchased from *Sigma Aldrich*) and sonicated for ~10 minutes. This sample is referred to as **I** from here onwards. Paramagnetic TEMPOL radicals were chosen to most efficiently polarize $^1$H spins under *d*DNP conditions. A 100 $\mu$L volume of **I** was pipetted into a Kel-F sample cup and inserted into a 7.05 T prototype *Bruker Biospin* polarizer equipped with a specialized *d*DNP probe and running *TopSpin 3.7* software. The sample temperature was reduced to 1.2 K by submerging the sample in liquid helium and reducing the pressure of the variable temperature insert (VTI) towards ~0.7 mbar.

## 2.2 Dynamic Nuclear Polarization

The sample was polarized by applying microwave irradiation at 197.648 GHz (positive lobe of the EPR line) with triangular frequency modulation of amplitude $\Delta f_{mw}$ = 120 MHz (Bornet et al., 2014) and rate $f_{mod}$ = 0.5 kHz at a power of c.a. 100 mW, which were optimized prior to commencing experiments to achieve the best possible level of $^1$H polarization. Microwave gating was employed shortly before and during *d*DNP transfer experiments to allow the electron spin ensemble to return to a highly polarized state, which happens on the timescale of the longitudinal electron relaxation time (typically $T_{1e}$ = 100 ms with $P_e$ = 99.93% under *d*DNP conditions) (Bornet et al., 2016). Consequently, the $^1$H and $^{13}$C relaxation times in the presence of a *rf*-field are extended by orders of magnitude, allowing spin-locking *rf*-pulses to be much longer which significantly increases the efficiency of nuclear polarization transfer.

## 2.3 Pulse Sequences

In 1967 Jeener and Broekaert established the original *rf*-pulse sequence for creating and observing dipolar order in the solid-state (Jeener and Broekaert, 1967). Since then, other *rf*-pulse sequences have been proposed in the literature, usually with improved efficiency (Jeener et al., 1965; Redfield, 1969; Kunitomo et al., 1974; Demco et al., 1975; Emid et al., 1980; Vieth and Yannoni, 1993; Zhang et al., 1993; Kurur and Bodenhausen, 1995; Lee and Khitrin, 2008; Khitrin et al., 2011; Vinther et al., 2019). Herein, we are most interested in the *rf*-pulse sequence introduced by Vieth and Yannoni (Vieth and Yannoni, 1993) which is particularly simple, easily generates proton dipolar order and allows subsequent conversion to $^{13}$C polarization. Figure 1 shows this sequence adapted for our *d*DNP experiments. An electron-nuclear variant of this *rf*-pulse sequence has also been developed (Macho et al., 1991; Buntkowsky et al., 1991).

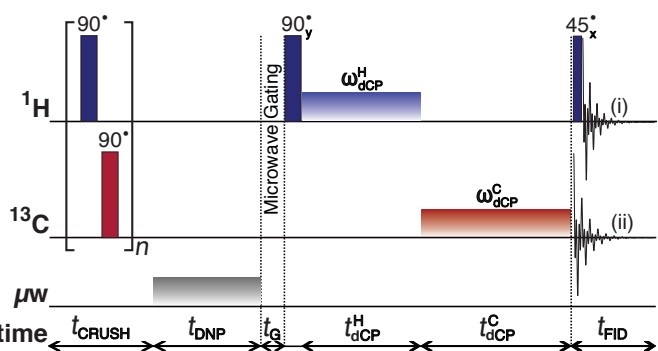

**Figure 1: Schematic representation of the *d*CP *rf*-pulse sequence used for preparing and monitoring $^1$H-$^1$H dipolar order in I, and the conversion to $^{13}$C transverse magnetization. The experiments used the following parameters, chosen to maximize magnetization-dipolar order interconversion: $n$ = 250; $t_{DNP}$**

= 5 s; $t_G$ = 0.5 s; $\omega_{dCP}^H/2\pi$ = 16.4 kHz; $t_{dCP}^H$ = 25 $\mu$s; $\omega_{dCP}^C/2\pi$ = 13.2 kHz; $t_{dCP}^C$ = 39 ms. The $^1$H and $^{13}$C spin-locking *rf*-pulses have phase $x$. The $\pi/2$ crusher
*rf*-pulses use a thirteen-step phase cycle to remove residual magnetization at the beginning of each experiment: {0, $\pi/18$, $5\pi/18$, $\pi/2$, $4\pi/9$, $5\pi/18$, $8\pi/9$,
$\pi$, $10\pi/9$, $13\pi/9$, $\pi/18$, $5\pi/3$, $35\pi/18$}. The resonance offset was placed at the centre of the $^1$H and $^{13}$C NMR peaks.
The *d*CP *rf*-pulse sequence operates as follows:
(*i*) A crusher sequence of 90° *rf*-pulses with alternating phases separated by a short delay (typically 11 ms) repeated *n* times
(typically *n* = 250) kills residual magnetization on both *rf*-channels;
(*ii*) The microwave source becomes active for a time $t_{DNP}$ during which $^1$H DNP builds-up;
(*iii*) The microwave source is deactivated and a delay of duration $t_G$ = 0.5 s occurs before the next step, thus permitting the
electron spins to relax to their highly polarized thermal equilibrium state (Bornet et al., 2016);
(*iv*) A $^1$H 90° *rf*-pulse followed by a $\pi/2$ phase-shifted spin-locking $^1$H *rf*-pulse of amplitude $\omega_{dCP}^H$ and length $t_{dCP}^H$ converts $^1$H
Zeeman polarization into $^1$H-$^1$H dipolar order;
(*v*) A $^{13}$C square *rf*-pulse of amplitude $\omega_{dCP}^C$ and length $t_{dCP}^C$ presumably converts the $^1$H-$^1$H dipolar order into $^{13}$C transverse
magnetization.
The NMR signal can be detected by using either: (*i*) a $^1$H 45° *rf*-pulse followed by $^1$H FID acquisition to monitor the remaining
proton dipolar order; or (*ii*) $^{13}$C FID detection to observe the converted magnetization, see Figure 1.
The *d*CP *rf*-pulse sequence can be used in several variants:
*Variant #1*: Efficiency of $^1$H-$^1$H dipolar order preparation.
(*a*) $^1$H observation by fixing $t_{dCP}^C$ = 0 ms and varying $\omega_{dCP}^H$ and $t_{dCP}^H$ (Figure 2a);
(*b*): $^{13}$C observation by fixing $t_{dCP}^C$ and $\omega_{dCP}^C$ (typically to an optimal value) and varying $\omega_{dCP}^H$ and $t_{dCP}^H$ (Figure 2c).
*Variant #2*: Efficiency of $^1$H-$^1$H dipolar order conversion to $^{13}$C magnetization.
(*a*): $^{13}$C observation by fixing $\omega_{dCP}^H$ and $t_{dCP}^H$ (typically to an optimal value) and varying $\omega_{dCP}^C$ and $t_{dCP}^C$ (Figure 3a);
(*b*): $^1$H observation by fixing $\omega_{dCP}^H$ and $t_{dCP}^H$ (typically to an optimal value) and varying $\omega_{dCP}^C$ and $t_{dCP}^C$ (Figure 4a).
The amplitudes of the $^1$H and $^{13}$C *d*CP *rf*-pulses ($\omega_{dCP}^H$ and $\omega_{dCP}^C$, respectively) were optimized iteratively until the intensity of
the resulting NMR signals could not be improved further, see the Electronic Supplementary Material (ESM) for more details.
In the case of proton *rf*-channel acquisition, data points were acquired with a two-step phase cycle, in which the phase of the
$90_y$ *rf*-pulse and the digitizer were simultaneously changed by 180° in successive transients, to remove spurious signals generated
by longitudinal magnetization accrued during the *d*CP *rf*-pulses. A dispersive lineshape was observed as a result of the phase cycle,
which is characteristic of dipolar spin order. The resulting $^1$H NMR spectrum was phase corrected to yield an absorptive lineshape.
**3 Results**
**3.1 $^1$H-$^1$H Dipolar Order Preparation**
$^1$H monitored optimization for the generation of $^1$H-$^1$H dipolar order as a function of the *d*CP $^1$H *rf*-pulse duration $t_{dCP}^H$ was
performed by using *variant #1a* of the *d*CP sequence shown in Figure 2a. Experimental results demonstrating the preparation of
$^1$H-$^1$H dipolar order under *variant #1a* of the *d*CP sequence are shown in Figure 2b. The integrals plotted were acquired directly
on the $^1$H *rf*-channel using $\omega_{dCP}^H/2\pi$ = 16.4 kHz either with or without microwave gating (black circles and grey squares,
respectively). In both cases, the NMR signal grows until a maximum signal intensity, which corresponds to the optimal preparation
of proton dipolar order, is reached at $t_{dCP}^H \simeq$ 25 $\mu$s, after which the signal decays towards a stable plateau on a longer timescale.
However, in the case that microwave gating is removed, the signal intensity is reduced. This is due to depolarization (microwave
saturation) of the electron spins, resulting in a detrimental enhancement of the paramagnetic relaxation contribution to nuclear spin

relaxation. These results suggest that microwave gating improves the conversion of $^{1}$H magnetization to $^{1}$H-$^{1}$H dipolar order by a factor of at least ~1.6.

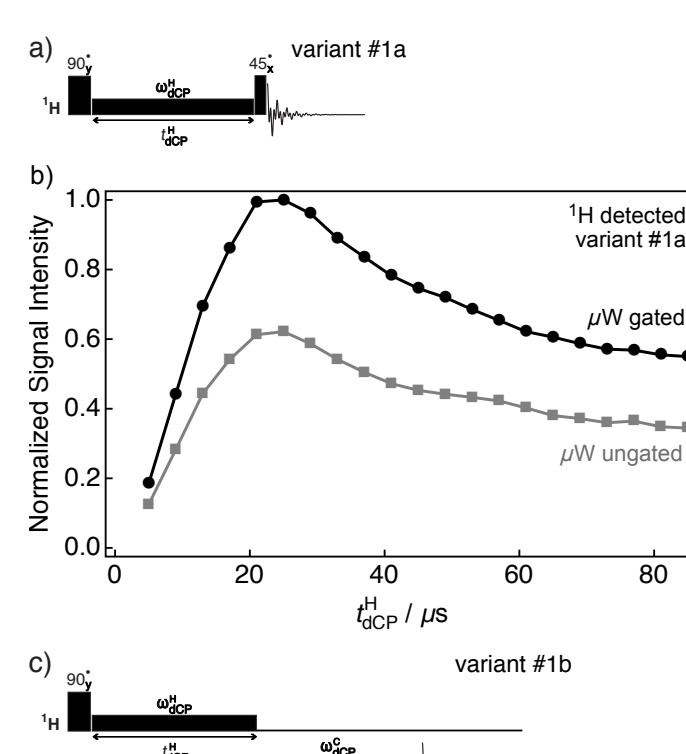

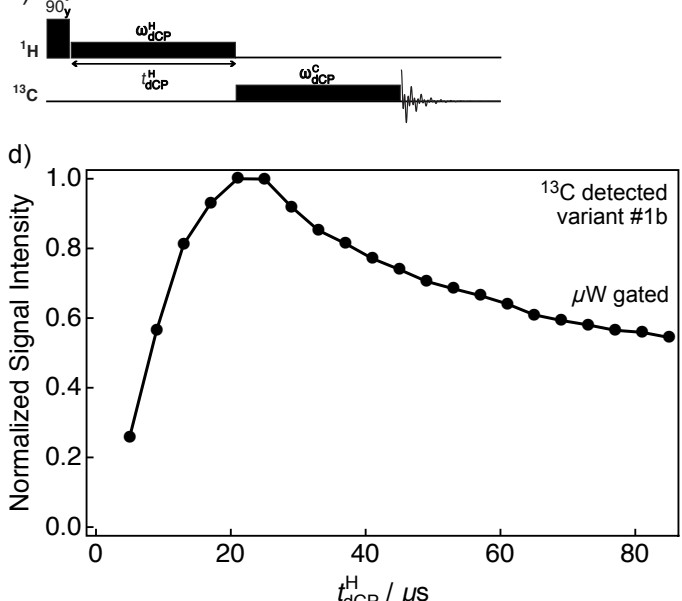

**Figure 2: Simplified schematic representations of (a) *variant #1a* and (c) *variant #1b* of the *d*CP *rf*-pulse sequence. Experimental (b) $^{1}$H and (d) $^{13}$C NMR signal intensities of I as a function of the $^{1}$H *d*CP *rf*-pulse duration $t_{dCP}^{H}$ acquired at 7.05 T ($^{1}$H nuclear Larmor frequency = 300.13 MHz, $^{13}$C nuclear Larmor frequency = 75.47 MHz) and 1.2 K. The experiments in (b) were acquired with two transients per data point, whilst the experiments in (d) were acquired with a single transient per data point. The traces have the same overall form, and plateau over a period of 200 $\mu$s (data not shown).**

$^{13}$C monitored optimization for the build-up of $^{1}$H-$^{1}$H dipolar order was performed by using *variant #1b* of the *d*CP *rf*-pulse sequence demonstrated in Figure 2c. In Figure 2d the experimental integrals are plotted against the *d*CP $^{1}$H *rf*-pulse duration $t_{dCP}^{H}$ and were acquired on the $^{13}$C *rf*-channel with $\omega_{dCP}^{H}/2\pi = 16.4$ kHz, $\omega_{dCP}^{C}/2\pi = 13.2$ kHz and $t_{dCP}^{C} = 39$ ms (black circles). It is important to note that the maximum is identical whether the NMR signal is observed on the $^{1}$H *rf*-channel by using *variant #1a* or on the $^{13}$C *rf*-channel by using *variant #1b*, and more generally that the two traces have the same shape and optimum. This shows that $^{13}$C transverse magnetization from *d*CP is proportional to the $^{1}$H-$^{1}$H dipolar order initially prepared.

## 3.2 $^1$H-$^{13}$C Polarization Transfer

Figure 3b shows how $^{13}$C magnetization is built-up by employing *variant #2a* the *d*CP *rf*-pulse sequence, see Figure 3a. The experimental integrals of the $^{13}$C signal are plotted against the $^{13}$C *d*CP *rf*-pulse duration $t_{\text{dCP}}^{\text{C}}$ with (black circles) and without (grey squares) microwave gating.

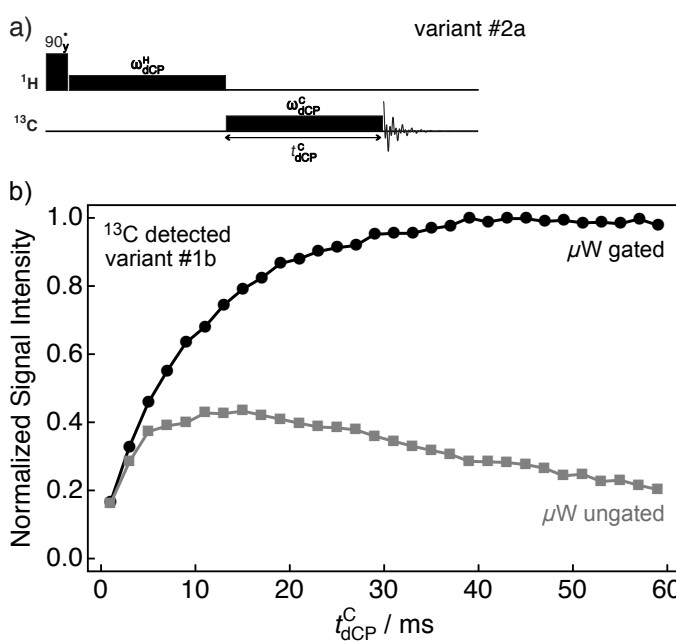

Figure 3: (a) Simplified schematic representation of *variant #2a* of the *d*CP *rf*-pulse sequence. (b) Experimental $^{13}$C NMR signal intensity of I as a function of the *d*CP *rf*-pulse duration $t_{\text{dCP}}^{\text{C}}$ acquired at 7.05 T ($^1$H nuclear Larmor frequency = 300.13 MHz, $^{13}$C nuclear Larmor frequency = 75.47 MHz) and 1.2 K with a single transient per data point.

The black trace corresponds to the growth of the $^{13}$C NMR signal. A maximum is reached at $t_{\text{dCP}}^{\text{C}} \simeq 39$ ms, with $\omega_{\text{dCP}}^{\text{C}} = 13.2$ kHz. The polarization transfer efficiency is relatively robust with respect to the amplitude of the $^{13}$C *d*CP *rf*-pulse $\omega_{\text{dCP}}^{\text{C}}$, see the ESM for more details. A wildly different behaviour is observed in the case where the microwave source is not gated. In this instance, a maximum signal intensity occurs at $t_{\text{dCP}}^{\text{C}} \simeq 15$ ms, with the detectable $^{13}$C signal decreasing past this point. The ratio between the maximum data points is ~2.3, and indicates a large $^{13}$C enhancement afforded by microwave gating.

It is worth noting that the duration of the $^{13}$C *d*CP *rf*-pulse is considerably longer, more than three orders of magnitude, than the $^1$H *d*CP *rf*-pulse lengths. Reasons for this are examined in the discussion section below.

Figure 4b details how in *variant #2b* of the *d*CP *rf*-pulse sequence (Figure 4a) the $^1$H NMR signal vanishes as the $^{13}$C *d*CP *rf*-pulse generates $^{13}$C transverse magnetization. The experimental integrals of the $^1$H detected NMR signals are plotted against the $^{13}$C *d*CP *rf*-pulse duration $t_{\text{dCP}}^{\text{C}}$ with $\omega_{\text{dCP}}^{\text{C}} = 0$ kHz (black open circles) and $\omega_{\text{dCP}}^{\text{C}} = 13.2$ kHz (black circles) both with microwave gating, and with $\omega_{\text{dCP}}^{\text{C}} = 13.2$ kHz (grey squares) without microwave gating.

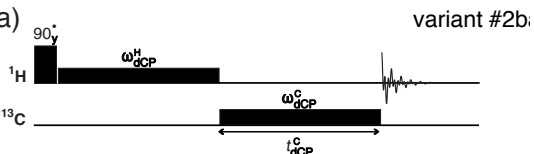

b)

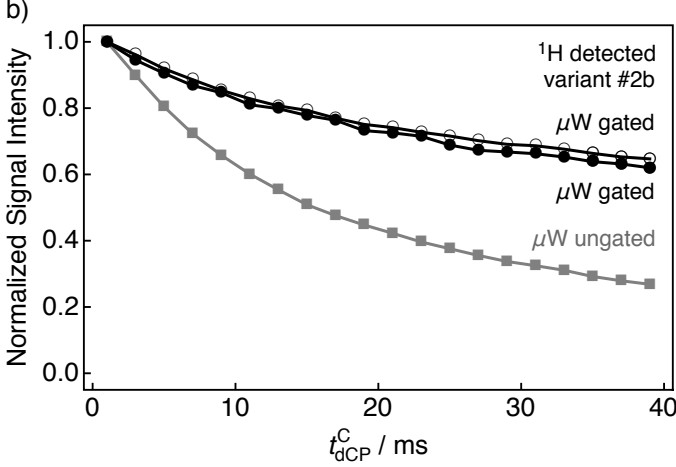

Figure 4: (a) Simplified schematic representation of *variant #2b* of the *d*CP *rf*-pulse sequence. (b) Experimental [1]H NMR signal intensity of I as a function of the [13]C *d*CP *rf*-pulse duration $t_{dCP}^C$ acquired at 7.05 T ([1]H nuclear Larmor frequency = 300.13 MHz, [13]C nuclear Larmor frequency = 75.47 MHz) and 1.2 K with two transients per data point. The experimental traces were recorded by using the following amplitudes for the [13]C *d*CP *rf*-pulse $\omega_{dCP}^C$: Black open circles: $\omega_{dCP}^C = 0$ kHz; Black filled circles: $\omega_{dCP}^C = 13.2$ kHz; Grey squares: $\omega_{dCP}^C = 13.2$ kHz. All signal amplitudes were normalized to the first data point.

The curves show how [1]H-[1]H dipolar order decays towards thermal equilibrium mainly through relaxation and is not significantly affected by the presence of the [13]C *d*CP *rf*-pulse generating [13]C magnetization. The difference between the two black traces might however indicate the quantity of [1]H-[1]H dipolar order converted into [13]C magnetization. The small difference is just a few percent, indicating that only a very small portion of the [1]H-[1]H dipolar order might be used (and be useful) to produce hyperpolarized [13]C magnetization. This could be explained by the large excess of [1]H spins compared with [13]C spins in our sample (a factor of ~6.2). A lack of microwave gating (grey squares) significantly compromises the generation of [13]C polarization, as seen in Figure 3b.

## 3.3 Comparison to Cross-Polarization

The performance efficiency of the *d*CP *rf*-pulse sequence was compared to a traditional CP experiment (Hartmann and Hahn, 1962; Pines et al., 1972; Perez Linde, 2009; Jannin et al., 2011; Bornet et al., 2012; Batel et al., 2012; Bornet et al., 2013; Vuichoud et al., 2016; Cavaillès et al., 2018), which is described in the ESM along with a *rf*-pulse sequence diagram and all optimized parameters. Experiments employed 640 s of direct [1]H DNP at 1.2 K prior to polarization transfer to the [13]C heteronucleus.

The power requirements for polarization transfer are dependent upon the *rf*-pulse sequence used and the capabilities of the *d*DNP probe. In general, the peak power for the [13]C *d*CP *rf*-pulse is ~5.4 times lower than required for CP. However, the [13]C *d*CP *rf*-pulse is active for a duration ~5.6 times longer than that of CP, and hence the overall deposited *rf*-pulse energy is approximately the same for both *rf*-pulse sequences. Notwithstanding, the moderately lower [13]C *d*CP *rf*-pulse power is highly advantageous, e.g. decreased likelihood of probe arcing events within the superfluid helium bath. The benefit of employing the *d*CP *rf*-pulse sequence becomes even more apparent when examining the proton *rf*-pulse durations needed for [1]H-[13]C polarization transfer. Although the peak powers of both *rf*-pulse sequences are similar, the duration of the [1]H *d*CP *rf*-pulse is a factor of 280 times shorter than that recommended for adequate CP. This is advantageous in the case that the $B_1$-field produced by the *d*DNP probe is weak (e.g. due to large sample constraints) or is unstable at higher [1]H *rf*-pulse powers for sufficiently long durations.

The CP *rf*-pulse sequence achieved a [13]C polarization level of $P(^{13}C) \simeq 20.4\%$ after a single CP contact. [13]C polarization levels in excess of 60% are anticipated by using a multiple CP contact approach (Perez Linde, 2009; Jannin et al., 2011; Bornet et al., 2012; Batel et al., 2012; Bornet et al., 2013; Vuichoud et al., 2016; Cavaillès et al., 2018). In comparison, the integral of the *d*CP-filtered NMR signal maximum is scaled by a factor of ~0.43, indicating a [13]C polarization of $P(^{13}C) \simeq 8.7\%$. This is consistent

with previous results reported in the literature (Perez Linde, 2009; Vinther et al., 2019). Strategies to further improve the *d*CP
efficiency are presented in the discussion section.

**4 Discussion**

The results presented in Figure 2b and Figure 2d show how the achieved $^{13}$C polarization is directly proportional to the quantity of
$^1$H-$^1$H dipolar order initially prepared by the $^1$H *d*CP *rf*-pulse. However, even if the $^{13}$C polarization closely follows the shape of
the proton dipolar order creation profile, this does not constitute irrefutable proof that the $^{13}$C polarization originates from the
proton dipolar order reservoir itself. Other, more-complex forms of nuclear spin order might be involved. Moreover, it is feasible
that an intermediate reservoir exists, such as non-Zeeman spin order of the $^{13}$C heteronucleus.
As seen in Figure 3b, it is interesting to note that the duration of the $^{13}$C *d*CP *rf*-pulse is considerably longer, more than three
orders of magnitude, than the $^1$H *d*CP *rf*-pulse duration. The reason is the relative sizes of the dipolar couplings which control the
preparation and transfer processes of $^1$H-$^1$H dipolar order. The generation of dipolar order involves only proton spins, which possess
a magnetogyric ratio ~4 times greater than for $^{13}$C spins and consequently larger dipolar couplings, which scale as the product of
the magnetogyric ratios for the two spins involved. This results in a short time to convert $^1$H magnetization to $^1$H-$^1$H dipolar order.
Conversely, the supposed transfer of $^1$H-$^1$H dipolar order to $^{13}$C nuclei would certainly demand $^1$H-$^{13}$C dipolar couplings.
The duration of the $^{13}$C *d*CP *rf*-pulse is a factor of ~5.6 longer than required for optimized conventional CP (see the ESM for
more details). The extended duration of the $^{13}$C *d*CP *rf*-pulse could be conceivably explained by assuming that the $^1$H spins closest
to the $^{13}$C spin do not participate in the polarization transfer process since the $^1$H-$^1$H dipolar order preparation is perturbed by the
presence of the $^{13}$C spin during the $^1$H *d*CP *rf*-pulse. It is also possible that two dipolar coupled protons are separated by a difference
in chemical shift which matches the frequency of a $^{13}$C spin the rotating frame allowing an exchange of energy. Such events are
similar to the cross-effect in DNP (Kessenikh et al., 1963) but are likely to be of lower probability, leading to an increased $^{13}$C *d*CP
*rf*-pulse duration.
Not only is the polarization transfer process long, but it is also weaker than what is usually realized with optimized CP, since
we obtain $P(^{13}\mathrm{C}) \simeq 8.7\%$ rather than $P(^{13}\mathrm{C}) \simeq 20.4\%$ in a single CP step on the same sample. Although the amplitude $\omega_{\mathrm{dCP}}^{\mathrm{H}}$ and
duration $t_{\mathrm{dCP}}^{\mathrm{H}}$ of the proton dipolar order creation *rf*-pulse were carefully optimized before experimental implementation, we
nevertheless believe there is still room for improvement in preparing high quantities of proton dipolar order. The performance of
the *d*CP *rf*-pulse sequence could be enhanced by adopting the following strategies: (*i*) employing shaped *rf*-pulses; (*ii*)
implementing a multiple *d*CP transfer approach; (*iii*) optimizing the protonation level of the DNP glassing solution; (*iv*) exploiting
deuterated molecular derivatives; (*v*) avoiding large quantities of methyl groups which may act as dipolar order relaxation sinks
due to their inherent rotation (which remains present at liquid helium temperature); and (*vi*) changing the molecule [1-$^{13}$C]sodium
acetate for another spin system with different $^1$H-$^{13}$C coupling strengths (e.g. simply using [2-$^{13}$C]sodium acetate).
Today's performances on our current 'standard' DNP sample are rather poor compared to CP, however, there are reasons to
think that further improvements through advanced *rf*-pulse schemes and revised sample formulations will be possible in the future,
and that *d*CP may become a viable alternative to CP. This will be particularly relevant to the cases of: (*i*) issues related to probe
arcing in the superfluid helium bath which precludes the use of conventional CP experiments; (*ii*) increased sample volumes, e.g.
in human applications; and (*iii*) hyperpolarization of insensitive nuclear spins, e.g. $^{89}$Y nuclei cannot be polarized easily via
traditional CP experiments due to unfeasible CP matching conditions on the heteronuclear *rf*-channel. Other alternatives to the CP
approach also exist but are theoretically less efficient, such as low magnetic field nuclear thermal mixing (Gadian et al., 2012)
which relies on energy conserving mutual spin-flips in overlapping NMR lineshapes to polarize heteronuclei in solid samples (Peat
et al., 2016).

## 5 Conclusions

$^1$H$\rightarrow$$^{13}$C polarization transfer occurs by employing *rf*-pulse methods which operate under *d*DNP conditions. This supposedly involves an intermediate reservoir of dipolar order, which governs the polarization transfer process. The spin dynamics of dipolar order mediated cross-polarization (*d*CP) were found to significantly depend on the presence of microwave gating. A maximum $^{13}$C polarization of ~8.7% was observed after ~10 minutes of microwave irradiation and a lone polarization step, which corresponds to a *d*CP polarization transfer efficiency of ~0.43 with respect to optimized conventional CP. These results are promising for future applications of polarization conversion methods in the context of low power $^1$H$\rightarrow$X polarization transfer to insensitive nuclei (in particular for very low magnetogyric ratios), with minimized probe arcing and potentially large sample volumes, paving the way to the use of $^1$H$\rightarrow$X polarization transfer in clinical (human-dose) contexts.

## Acknowledgements

The authors gratefully acknowledge *Bruker Biospin* for providing the prototype *d*DNP polarizer, and particularly Dmitry Eshchenko, Roberto Melzi, Marc Rossire, Marco Sacher and James Kempf for scientific and technical support. The authors additionally acknowledge Gerd Buntkowsky (Technische Universitat Darmstadt) who kindly communicated data associated with prior publications to us; Burkhard Luy (Karlsruhe Institute of Technology) for enlightening discussions; Catherine Jose and Christophe Pages for use of the ISA Prototype Service; and Stéphane Martinez of the UCBL mechanical workshop for machining parts of the experimental apparatus.

## Financial Support

This research was supported by ENS-Lyon, the French CNRS, Lyon 1 University, the European Research Council under the European Union's Horizon 2020 research and innovation program (ERC Grant Agreements No. 714519 / HP4all and Marie Skłodowska-Curie Grant Agreement No. 766402 / ZULF).

## Author Contributions

SJE performed experiments and co-wrote the manuscript, SFC/QC/OC/AB performed experiments, MC built parts of the experimental apparatus, and SJ conceived the idea and co-wrote the manuscript.

## Competing Interests

The authors declare no competing interests.

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
