# Peer review of "Dipolar Order Mediated ${}^{1}H \rightarrow {}^{13}C$ Cross-Polarization for Dissolution-1 Dynamic Nuclear Polarization 2"

_Magnetic Resonance, 2020_

## Short Comment (SC1) · 11 Mar 2020

This manuscript is interesting as it aims at the development of an alternative pathway (transfer via 1H) to enhance 13C polarization under dDNP conditions. The proposed 1H-13C transfer mechanism is different from that in the conventional Hartmann-Hahn CP, which is based on the idea of spin lock on both the channels. The new RF scheme requires non-simultaneous RF irradiation on the two channels.

Although the optimum RF power is not lower (compared to that in conventional CP), the duration of RF on 1H is significantly reduced. However that on 13C channel becomes significantly longer and the transfer efficiency also decreases by a factor of 2.

Overall, the new approach might still be better than Hartmann-Hahn CP for technical reasons as the authors claim. I have a few questions regarding the 1H-13C transfer mechanism

1. On page 2 line 31, the authors talk about conversion of 1H Zeeman polarization to I1z.I2z "dipolar order" using RF. How can the proposed rf on 1H convert Iz to I1z.I2z term, can authors provide some more insights?

2. For transfer of polarization from I to S, there has to be an effective ZQ or DQ IS dipolar Hamiltonian. Since there are RF pulses on both the channels, such Hamiltonian terms can be generated. But they seem to be of higher order perturbation term in the Hamiltonian. Maybe that's the reason why transfer rate is very slow. Can authors shed some light on this?

3. For the given rf scheme, generating a purely ZQ or a purely DQ (IS) dipolar Hamiltonian might be challenging. This in turn may lead to phase distortion of the 13C signal if there are multiple 13C resonances. Can authors provide some 13C spectra?

4. Since the method is based on 1H "dipolar order", what kind of spin-system is required for it to be efficient. Can the transfer mechanism be elucidated using a simple three-spin 1H-1H-13C model? How would 1H concentration in glassy matrix influence this transfer?

---

## Referee Comment (RC1) · Anonymous Referee #1 · 26 Mar 2020

The manuscript entitled "Dipolar Order Mediated 1H→13C Cross-Polarization for Dissolution-Dynamic Nuclear Polarization" by Stuart Elliott et al. entails a discussion of an alternative way, apart from Hartmann-Hahn cross polarization method, to harness the high nuclear polarization of hyperpolarized 1H spins and transfer it to 13C spins via simpler, low-power, and non-synchronized 1H and 13C RF pulses.

In my opinion, the experimental demonstration of 1H to 13C polarization transfer mediated by dipolar order is certainly a welcome addition to the technical developments in dissolution DNP, in pursuit of simpler alternative to DNP cross polarization in terms of RF hardware and pulses. One of the main advantages of this reported technique is the

use of non-simultaneous 1H and 13C RF pulses in the DNP polarization transfer. This reported technique also opens up an avenue for polarizing larger DNP sample volumes with minimal probe arcing. For these and other reasons, I believe that this manuscript is significant in terms of scientific content and it brings some new technical insights to the magnetic resonance community, in particular to the rapidly growing dissolution DNP field. Therefore, I would like to recommend publication of this manuscript with minor revision addressing the following suggestions and comments:

(1) Page 1: In the title, should it be "Dipolar Order-Mediated..." with the dash? (2) Page 1: lines 37 and 43–please spell out "typ." to typically. (3) Page 4: line 26–same comment as #2. (4) page 4: line 29–should be "the microwave is deactivated" (5) Page 7: Figure 4 caption, line 17–"nuclear Larmor frequency" was used twice; I suggest to use symbol omega or make it concise.

(6) The authors mentioned that this dipolar order-mediated CP technique ($\sim$8.7%) is only about a half as efficient compared to the conventional CP-DNP technique ($\sim$20%) in terms of final 13C DNP_enhanced polarization obtained. Do the authors have 13C polarization value for direct 13C polarization (without CP or dipolar order CP) of this sample?

(7) I assume these numbers (8.7% for dipolar-order CP, 20% for conventional CP) are solid-state 13C polarizations. Do the authors have liquid-state 13C polarization numbers (post dissolution)? These are not a requirement for this paper, but I think it would be good to report them if the data are available.

(8) Obviously there's a lot of optimization to be done here in this preliminary technical report especially with DNP sample optimization. Can the authors expound on the possible effects of the efficiency of dipolar order-mediated CP if the 1H spin density is increased or decreased in the glassing matrix?

(9) The supplementary material is appropriate.

Overall, I believe that this is a significant manuscript that merits publication in Magnetic Resonance pending this minor revision.

---

## Referee Comment (RC2) · Anonymous Referee #2 · 5 Apr 2020

Following the introduction of dissolution-DNP (dDNP) by Golman and Ardenkjær-Larsen, there have been discussions of approaches to shorten the hours long times required for the e-âđİ13C polarization transfer process. This step is limited by the slow 13C-13C spin diffusion process. Improvements are impeded by the fact that GE/Oxford/etc. does not permit investigators to modify their dDNP equipment – for example, by adding an 1H tuning circuit to the single resonance 13C circuit present in their probes.

In addition, it has been known since the 1970's that 1H's polarize much more rapidly than low-ðİŽĎ species such as 13C or 15N. For example, Hartmann, et al. (Nuclear

Instruments and Methods 106, 9-12 (1973)) showed that 1H's in alcohol samples at 1K and 5 T could be polarized in <2 minutes to levels of 35-70

This paper by Elliot, et al is a description of some of the approaches to implement the 1HâđĮ13C transfers that utilize low powers to avoid arcing in the helium atmosphere. The schemes are based on: (i) Less (or low) rf-power; (ii) less overall rf-energy; (iii) simple rf-pulse shapes; and (iv) no synchronized of the 1H and 13C rf-irradiation.

The transfer schemes are designed to take advantage of the terms in the expansion of the density matrix that go as $I_{izx}I_{jz}$, a dipolar order term that becomes important at low temperatures. The approaches use a gated microwave field and the di j

The paper is largely okay as written. However, I would suggest that the authors consider the following to improve the scholarship of the paper.

(1) I would include the reference to Hartmann (1973) above that, as far as I am aware, was the first to report the short polarization times of 1H at 1-2 K. The dDNP community pretty much ignores the extensive DNP physics literature from the 1960-2000 era and starts by quoting Golman and Ardenkjær-Larsen in 2003. In fact, I would suggest that they do a literature search to see if others have also reported these short polarization times.

(2) They also mention that the microwaves are gated and swept with a triangular frequency modulation. It would be good to discuss this in more detail. Why was the width of 120 MHz and a rate of 500 Hz chosen ? There are AWG's available these days that can easily produce more interesting waveforms. Have any of these been introduced into the experiment ? For example, the waveform could be adiabatic which might be more efficient that a simple triangular waveform.

(3) Why was the TEMPO concentration set at 50 mM ? This is about 3 times that used in MAS experiments and x3 the 15 mM concentration of trityl often employed by Ardenkjær-Larsen and coworkers in their experiments. Does the higher concentration

lead to the shorter polarization periods ? A fair comparison of polarization levels and build up times and would compare 15mM trityl to 15 mM TEMPO.

(4) Is the transfer mechanism established to be thermal mixing or the cross effect ? Although this is not the focus of the paper, it should be mentioned and discussed at least briefly. If the cross effect is involved then why doesn't the dDNP community use nitroxide biradicals as polarizing agents. Again this could be briefly discussed.

---

## Author Comment (AC1) · 12 Apr 2020

This manuscript is interesting as it aims at the development of an alternative pathway (transfer via [1]H) to enhance [13]C polarization under *d*DNP conditions. The proposed [1]H-[13]C transfer mechanism is different from that in the conventional Hartmann-Hahn CP, which is based on the idea of spin-locking on both the channels. The new RF scheme requires non-simultaneous RF irradiation on the two channels.

Although the optimum RF power is not lower (compared to that in conventional CP), the duration of RF on [1]H is significantly reduced. However that on [13]C channel becomes significantly longer and the transfer efficiency also decreases by a factor of 2. Overall, the new approach might still be better than Hartmann-Hahn CP for technical reasons as the authors claim.

I have a few questions regarding the [1]H-[13]C transfer mechanism (*the authors responses are given in italics*):

1. On page 2 line 31, the authors talk about conversion of [1]H Zeeman polarization to $I_{1z}.I_{2z}$ "dipolar order" using RF. How can the proposed RF on [1]H convert $I_z$ to $I_{1z}.I_{2z}$ term, can authors provide some more insights?

*In the spin-lock frame, the magnetization shrinks to a value which is small relative to that produced by the static magnetic field (for the same sample) during the demagnetization process. This effect has been extensively covered in a number of other papers:*

J. Jeener and P. Broekaert, *Phys. Rev.*, **1967**, 157, 232-240.
H.-M. Vieth and C. S. Yannoni, *Chem. Phys. Lett.*, **1993**, 205, 153-156.
J. Jeener, R. Du Bois and P. Broekaert, *Phys. Rev.*, **1965**, 139, A1959-A1961.
A. G. Redfield, *Science*, **1969**, 164, 1015-1023.
J.-S. Lee and A. K. Khitrin, *J. Chem. Phys.*, **2008**, 128, 114504.

*However, this is not the main focus of our current paper. Clearly, our spin system of choice is much more complicated than those in the above references (due to the presence of paramagnetic radicals etc.). Multiple dipolar orders or higher spin orders could also exist, as eluded to in our current paper. We aim to analyse these processes in greater detail in our future works.*

2. For transfer of polarization from I to S, there has to be an effective ZQ or DQ IS dipolar Hamiltonian. Since there are RF pulses on both the channels, such Hamiltonian terms can be generated. But they seem to be of higher order perturbation term in the Hamiltonian. Maybe that's the reason why transfer rate is very slow. Can authors shed some light on this?

*We surmise that since the dCP transfer is slower than conventional cross-polarization it involves a different transfer mechanism. However, we do not have sufficient insights into the true polarization transfer mechanism for this sample under dDNP conditions at present.*

3. For the given RF scheme, generating a purely ZQ or a purely DQ (IS) dipolar Hamiltonian might be challenging. This in turn may lead to phase distortion of the [13]C signal if there are multiple [13]C resonances. Can authors provide some 13C spectra?

[Figure]

*The recorded [13]C NMR spectra do not show significant phase distortion and are detected with a good level of signal-to-noise. Please see the attached [13]C NMR spectrum. Additional [13]C NMR spectra will be provided in a following publication on a similar topic.*

4. Since the method is based on [1]H "dipolar order", what kind of spin-system is required for it to be efficient. Can the transfer mechanism be elucidated using a simple three-spin [1]H-[1]H-[13]C model? How would [1]H concentration in glassy matrix influence this transfer?

*We have started simulations on a simple 3-spin-1/2 [1]H-[1]H-[13]C model system. However, agreement between simulated and experimental data has not yet been reached. It is this fact which provides a hint to the authors that a similar reservoir of non-Zeeman spin order may be used instead. There are also preliminary data to support this*

conclusion. Increasing the proton concentration of the glassy matrix dramatically decreases the polarization transfer time and increases its efficiency.

---

## Author Comment (AC2) · 12 Apr 2020

The manuscript entitled "Dipolar Order Mediated $^1$H→$^{13}$C Cross-Polarization for Dissolution-Dynamic Nuclear Polarization" by Stuart J. Elliott et al. entails a discussion of an alternative way, apart from the Hartmann-Hahn cross-polarization method, to harness the high nuclear polarization of hyperpolarized $^1$H spins and transfer it to $^{13}$C spins via simpler, low-power and non-synchronized $^1$H and $^{13}$C *rf*-pulses.

In my opinion, the experimental demonstration of $^1$H to $^{13}$C polarization transfer mediated by dipolar order is certainly a welcome addition to the technical developments in (dissolution-dynamic nuclear polarization *d*DNP), in pursuit of simpler alternative to DNP cross polarization in terms of *rf*-hardware and *rf*-pulses. One of the main advantages of this reported technique is the use of non-simultaneous $^1$H and $^{13}$C *rf*-pulses in the DNP polarization transfer. This reported technique also opens up an avenue for polarizing larger DNP sample volumes with minimal probe arcing. For these and other reasons, I believe that this manuscript is significant in terms of scientific content and it brings some new technical insights to the magnetic resonance community, in particular to the rapidly growing *d*DNP field. Therefore, I would like to recommend publication of this manuscript with minor revision addressing the following suggestions and comments:

The author response is given in italics.

(Q1) Page 1: In the title, should it be "Dipolar Order-Mediated..." with the dash? (Q2) Page 1: lines 37 and 43– please spell out "typ." to typically. (Q3) Page 4: line 26–same comment as #2. (Q4) page 4: line 29–should be "the microwave is deactivated" (Q5) Page 7: Figure 4 caption, line 17–"nuclear Larmor frequency" was used twice; I suggest to use symbol omega or make it concise.

*(A1) The authors have changed the title. (A2,A3) These changes have been made throughout the manuscript. (A4) The spelling has been corrected. (A5) The authors will stick to the current notation in order to be consistent throughout the manuscript.*

(Q6) The authors mentioned that this dipolar order-mediated CP technique (~8.7%) is only about a half as efficient compared to the conventional CP-DNP technique (~20.4%) in terms of the final $^{13}$C DNP-enhanced polarization obtained. Do the authors have a $^{13}$C polarization value for direct $^{13}$C polarization (without CP or dipolar order CP) of this sample?

*(A6) The $^{13}$C nuclear polarization level for direct DNP was unfortunately not recorded for this sample because it is inefficient and displays a very long build-up time.*

(Q7) I assume these numbers (~8.7% for dipolar-order CP, ~20.4% for conventional CP) are solid-state $^{13}$C polarizations. Do the authors have liquid-state $^{13}$C polarization numbers (post dissolution)? These are not a requirement for this paper, but I think it would be good to report them if the data are available.

*(A7) The $^{13}$C nuclear polarization values presented were measured in the solid-state. Liquid state $^{13}$C polarization levels (post dissolution) could be measured in the future and be presented as part of a separate publication.*

(Q8) Obviously there's a lot of optimization to be done here in this preliminary technical report especially with DNP sample optimization. Can the authors expand on the possible effects of the efficiency of dipolar order-mediated CP if the $^1$H spin density is increased or decreased in the glassing matrix?

*(A8) Upon increasing the $^1$H spin density within the glassing matrix, an improvement is observed in the performance of the dCP rf-pulse sequence with respect to that of a sophisticated and high rf-power CP experiment.*

---

## Author Comment (AC3) · 12 Apr 2020

Following the introduction of dissolution-DNP ($d$DNP) by Golman and Ardenkjær-Larsen, there have been discussions of approaches to shorten the hours long times required for the $e^- \rightarrow {}^{13}C$ polarization transfer process. This step is limited by the slow ${}^{13}C$-${}^{13}C$ spin diffusion process. Improvements are impeded by the fact that *GE*/*Oxford*/etc. does not permit investigators to modify their $d$DNP equipment -- for example, by adding a ${}^1H$ tuning circuit to the single resonance ${}^{13}C$ circuit present in their probes.

In addition, it has been known since the 1970's that ${}^1H$'s polarize much more rapidly than low-$\gamma$ species such as ${}^{13}C$ or ${}^{15}N$. For example, Hartmann, et al. (*Nuclear Instruments and Methods*, **106**, 9-12 (1973)) showed that ${}^1H$'s in alcohol samples at 1 K and 5 T could be polarized in <2 minutes to levels of 35-70%.

This paper by Elliott, et al is a description of some of the approaches to implement the ${}^1H \rightarrow {}^{13}C$ transfers that utilize low powers to avoid arcing in the helium atmosphere. The schemes are based on: (*i*) less (or low) *rf*-power; (*ii*) less overall *rf*-energy; (*iii*) simple *rf*-pulse shapes; and (*iv*) no synchronized of the ${}^1H$ and ${}^{13}C$ *rf*-irradiation. The transfer schemes are designed to take advantage of the terms in the expansion of the density matrix that go as $I_{iz} \times I_{jz}$; a dipolar order term that becomes important at low temperatures. The approach uses a gated microwave field and then different approaches to transfer polarization from ${}^1H$ to ${}^{13}C$ in Na-acetate. The paper is largely okay as written. However, I would suggest that the authors consider the following to improve the scholarship of the paper.

The author response is given in italics.

(Q1) I would include the reference to Hartmann (1973) above that, as far as I am aware, was the first to report the short polarization times of ${}^1H$ at 1-2 K. The $d$DNP community pretty much ignores the extensive DNP physics literature from the 1960-2000 era and starts by quoting Golman and Ardenkjær-Larsen in 2003. In fact, I would suggest that they do a literature search to see if others have also reported these short polarization times.

*(A1) A reference to Hartmann's 1973 paper has now been included.*

(Q2) They also mention that the microwaves are gated and swept with a triangular frequency modulation. It would be good to discuss this in more detail. Why was the width of 120 MHz and a rate of 500 Hz chosen? There are AWG's available these days that can easily produce more interesting waveforms. Have any of these been introduced into the experiment? For example, the waveform could be adiabatic which might be more efficient that a simple triangular waveform.

*(A2) The width and rate of the microwave field was optimized in order to give the best ${}^1H$ polarization during active ${}^1H$ DNP. A sentence regarding this information has been added to the manuscript: "The sample was polarized by applying microwave irradiation at 197.648 GHz (positive lobe of the EPR line) with triangular frequency modulation of amplitude $\Delta f_{mw}$ = 120 MHz (Bornet et al., 2014) and rate $f_{mod}$ = 0.5 kHz at a power of c.a. 100 mW, which were optimized prior to commencing experiments to achieve the best possible level of ${}^1H$ polarization." Detailed information concerning microwave gating is given on page 3/line 100 of the manuscript. More sophisticated AWG's have not been introduced into the experiment at present.*

(Q3) Why was the TEMPO concentration set at 50 mM? This is about 3 times that used in MAS experiments and x3 the 15 mM concentration of trityl often employed by Ardenkjær-Larsen and coworkers in their experiments. Does the higher concentration lead to the shorter polarization periods? A fair comparison of polarization levels and build-up times and would compare 15mM trityl to 15 mM TEMPO.

*(A3) At present, a concentration of 50 mM of TEMPO radical is used as a "standard sample" within our laboratory, and it is true that the high radical concentration would lead to shorter polarization build-up times. For future experiments, in which the authors plan to dissolve and transfer the sample to a separate superconducting magnet for detection, we will likely use a lower radical concentration c.a. 25 mM TEMPO radical.*

(Q4) Is the transfer mechanism established to be thermal mixing or the cross effect? Although this is not the focus of the paper, it should be mentioned and discussed at least briefly. If the cross effect is involved then why doesn't the $d$DNP community use nitroxide biradicals as polarizing agents. Again this could be briefly discussed.

*(A4) For ${}^1H$ nuclei at 1.2 K and 7.05 T the electron-proton transfer will presumably occur through thermal mixing and/or cross-effect. So far, bi-radicals have not shown better performances than TEMPO(L) in our experimental conditions. We prefer to keep this very complicated discussion out of the paper.*